# Plants Metabolome Study: Emerging Tools and Techniques

**DOI:** 10.3390/plants10112409

**Published:** 2021-11-08

**Authors:** Manish Kumar Patel, Sonika Pandey, Manoj Kumar, Md Intesaful Haque, Sikander Pal, Narendra Singh Yadav

**Affiliations:** 1Department of Postharvest Science of Fresh Produce, Agricultural Research Organization, Volcani Center, Rishon LeZion 7505101, Israel; 2Independent Researcher, Civil Line, Fathepur 212601, India; sonikapandey14@gmail.com; 3Institute of Plant Sciences, Agricultural Research Organization, Volcani Center, Rishon LeZion 7505101, Israel; manojbiochem16@gmail.com; 4Fruit Tree Science Department, Newe Ya’ar Research Center, Agriculture Research Organization, Volcani Center, Ramat Yishay 3009500, Israel; intesafulhaque@gmail.com; 5Plant Physiology Laboratory, Department of Botany, University of Jammu, Jammu 180006, India; sikanderpal@jammuuniversity.ac.in; 6Department of Biological Sciences, University of Lethbridge, Lethbridge, AB T1K 3M4, Canada

**Keywords:** analytical tools, data analysis, genetically modified crops, mass spectrometry, metabolomics databases, metabolomics software tools, omics, plant biology

## Abstract

Metabolomics is now considered a wide-ranging, sensitive and practical approach to acquire useful information on the composition of a metabolite pool present in any organism, including plants. Investigating metabolomic regulation in plants is essential to understand their adaptation, acclimation and defense responses to environmental stresses through the production of numerous metabolites. Moreover, metabolomics can be easily applied for the phenotyping of plants; and thus, it has great potential to be used in genome editing programs to develop superior next-generation crops. This review describes the recent analytical tools and techniques available to study plants metabolome, along with their significance of sample preparation using targeted and non-targeted methods. Advanced analytical tools, like gas chromatography-mass spectrometry (GC-MS), liquid chromatography mass-spectroscopy (LC-MS), capillary electrophoresis-mass spectrometry (CE-MS), fourier transform ion cyclotron resonance-mass spectrometry (FTICR-MS) matrix-assisted laser desorption/ionization (MALDI), ion mobility spectrometry (IMS) and nuclear magnetic resonance (NMR) have speed up precise metabolic profiling in plants. Further, we provide a complete overview of bioinformatics tools and plant metabolome database that can be utilized to advance our knowledge to plant biology.

## 1. Metabolomics: Plant Biology Perspective

Metabolomics is one of the fastest developing and attractive disciplines of the omics field, with huge potential and prospects in crop improvement programs. It is vital to review the abiotic/biotic stress tolerances and metabolomics-assisted breeding of crop plants [1]. Recent metabolomics platforms play a crucial role in exploring unknown regulatory networks that control plant growth and development [1]. Further innovative metabolomics application, called ecological metabolomics, deals with studying the biochemical interactions among plants across different temporal and spatial networks [2]. It describes the biochemical nature of various vital ecological phenomena, such as the effects of parasite load, the incidence of disease, and infection. It also helps to decode the potential impact of biotic and abiotic stresses on any critical biochemical process through the detection of metabolites [1]. Modern metabolomics platforms are being exploited to explain complex biological pathways and explore hidden regulatory networks controlling crop growth and health.

The performance of metabolomics study relies on its methodologies and instruments to comprehensively identify and measure each metabolite [3]. The complexity of the various metabolic characteristics and molecular abundances makes metabolomics a challenging task. Metabolomics or metabolite profiling terms are alternatively used to define three types of approaches, such as untargeted metabolomics, targeted metabolomics, and semi-targeted metabolomics [4,5]. Several integrated technologies and methodologies such as mass spectrometry (MS) based methods, including gas chromatography-mass spectrometry (GC-MS), liquid chromatography mass-spectroscopy (LC-MS), capillary electrophoresis-mass spectrometry (CE-MS), fourier transform ion cyclotron resonance-mass spectrometry (FTICR-MS) matrix-assisted laser desorption/ionization (MALDI), ion mobility spectrometry (IMS) and nuclear magnetic resonance (NMR) are used for large-scale analysis of highly complex mixtures of plant extracts [6]. In fact, these analytical methods have shown their potential in many plant species, including halophytes, medicinal plants, and food crops such as *Salicornia brachiata*, *Cuminum cyminum*, *Plantago ovata*, *Solanum lycopersicum*, *Oryza sativa*, *Triticum aestivum,* and *Zea mays* [7,8,9,10,11,12,13,14] (Table 1). In the last decade, a significant rise in the use of integrated metabolomics analysis methods has been reported over individual analytical platforms, as the latter does not provide holistic aspects of a plant metabolome [3].

Since the beginning of the 21st century, major developments in various ‘omics’ fields, such as genomics, transcriptomics, proteomics, metabolomics, and phenomics, have been seen. The various omics platforms have an endless potential to enhance the current understanding of complex biological pathways, allowing us to develop new approaches for crops improvement [15]. Metabolomics is one of the most complex approaches among other omics approaches and has received attention in agriculture science, especially for plant selections in a molecular breeding program. Therefore, metabolomics is used to acquire a vast amount of useful knowledge by accurate and high throughput peak annotation through the snapshot of the plant metabolome for the novel genes and pathways elucidation [16]. The combination of metabolomic integrated with transcriptomic analysis was successfully used to find out several possible approaches such as breeding and genome editing involved in activating metabolic pathways and gene expression [17]. Nevertheless, plant metabolomics has become an effective tool for exploring different aspects of system biology, greatly expanding our knowledge of the metabolic and signaling pathways in plant growth, development, and response to stress for improving the quality and yield of crops [18]. This review describes the plant metabolome (primary and secondary metabolites), metabolomics in genetically modified (GM) crops, including different analytical techniques, bioinformatics tools, and plant metabolome database.

### 1.1. Primary Metabolites

Primary metabolites are essential for plant growth and development as they are involved in various physiological and biochemical processes [15]. Primary metabolites include different classes of metabolites such as sugars, fatty acids, and amino acids, serving as vital functions such as osmolytes and osmoprotectants in plants under biotic and abiotic stresses [4,19]. Lipidomics is the comprehensive analysis of lipids in a biological system, including quantification and metabolic pathways. Alteration in lipid metabolism and composition are linked to changes in plant growth, development, and responses to a variety of environmental stressors [20]. Lipidomics can be divided into shotgun and targeted analysis. Shotgun lipidomics identifies all lipid species in a sample without prior knowledge of their composition, whereas targeted lipidomics analyzes a specific group of lipids [21]. LC-MS has been used widely in both global and targeted lipidomics [22,23,24]. Lipidomics is also utilized to understand better the function of genes involved in lipid metabolism in transgenic plants and manipulate complex lipid metabolism to produce long-chain fatty acids, especially omega-3 species in plants [25]. Yu et al. [26] utilized lipidomics analysis based on high-throughput and high-sensitivity mass spectrometry to characterize membrane lipid responses, which also captures a variety of oxidized lipids.

The nutritional markers α-linolenic acid and linoleic acid were detected in the leaves of *P*. *ovata* [10]. Linoleic acid predominated in the husk of *P*. *ovata*, followed by oleic acid, palmitic acid, stearic acid, and cis-11,14-eicosadienoic acid [10]. Seed fatty acid composition analysis of the *Paeonia rockii*, *P. potaninii*, and *P. lutea* revealed that α-linolenic acid was the most abundant, followed by oleic and linoleic acids [27]. According to the fatty acid content, all halophytes (non-succulent, succulent and shrubby halophytes) are high in α-linolenic acid, followed by linolenic and palmitic acid [28]. Oil and oleic acid content increased, while palmitic and linolenic acid content decreased during seed development *Jatropha curcas* [29]. The total lipid and fatty acid levels were strongly linked with the different developmental stages of the *P*. *ovata* fruit, according to principal component analysis (PCA), and the heat map revealed the differential fatty acid composition [9].

The highest content of threonine followed by glutamic acid, tyrosine, and aspartic acid were quantified in *Amaranthus hypochondriacus* and it is notable that amino acids, glutamic acid, and aspartic acid were among the main contributors [30]. The content of histidine, isoleucine, leucine, threonine, and lysine in leaves was considerably higher than in seeds and husks of *P*. *ovata* [10]. Glucose-6-phosphate, xylose, 2-piperidine carboxylic acid, monoamidomalonic acid, tryptophan, phenylalanine, histidine and carbodiimide were found to be key metabolites play a vital role in the plant metabolism of *Fritillaria thunbergii* [31]. Furthermore, the amino acid profile of *Cuminum cyminum* plants revealed that the levels of most amino acids (except asparagine) increased in plants subjected to salinity stress when compared to control plants [8]. Under salinity stress, two varieties of *Cicer arietinum* (Genesis 836 and Rupali) showed increased levels of sugar alcohols, including galactitol, erythritol, arabitol, xylitol, mannitol, and inositol, showing the importance of these metabolites in salt tolerance [32]. Nitric oxide-induced accumulation of amino acids, sugars, polyols, organic acids, and but not fatty acids and lipids in *C. arietinum* [33] (Table 1).

**Table 1 plants-10-02409-t001:** Identification of key metabolites in various plant species using different analytical methods.

Plant Species	Class	Analytical Tools	Key Metabolites	Reference
**Primary metabolites**				
*Plantago ovata*	Fatty acids	GC-MS	α-linolenic acid, linoleic acid and palmitic acid	[10]
*P. ovata*	Fatty acids	GC-MS	Pentadecanoic acid, palmitic acid, heptadecanoic acid, stearic acid, oleic acid, linoleic acid, γ-linolenic acid and arachidic acid	[9]
*Jatropha curcas*	Fatty acids	GC	Oleic acid, palmitic acid and linolenic acid	[29]
*Paeonia rockii, P. potaninii, and P. lutea*	Fatty acids	GC-MS	α-linolenic acid, oleic acid and linoleic acid	[27]
*Cicer* *arietinum*	Fatty acids	GC-MS	Pentadecanoic acid, palmitic acid, palmitoleic acid, stearic acid, oleic acid, linoleic acid, α-linolenic acid and arachidic acid	[33]
*P. ovata*	Amino acids	HPLC	Isoleucine, threonine, leucine, histidine and lysine	[10]
*P. ovata*	Amino acids	HPLC	Aspartate, glutamine, glycine, alanine, arginine, serine, proline, isoleucine and methionine	[9]
*Fritillaria thunbergii*	Amino acids	GC-MS	Tryptophan, phenylalanine and histidine	[31]
*C.* *arietinum*	Amino acids	GC-MS	L-glutamic acid, L-tryptophan, phenylalanine, glycine, serine, L-threonine, L-valine, L-ornithine and L-proline	[33]
*C.* *arietinum*	Sugars and Sugar alcohols	GC-MS	Sucrose, cellobiose, galactose, methylgalactoside, *myo*-inositol	[33]
*C.* *arietinum*	Sugar alcohols	GC-QqQ-MS	Galactitol, erythritol, arabitol, xylitol, mannitol and inositol	[32]
**Secondary metabolites**				
*Beta* *vulgaris*	Terpenes	HPLC-MS	Oleanolic acid, hederagenin, akebonoic acid and gypsogenin	[34]
*Ocimum gratissimum*	Terpenes	GC-MS	m-chavicol, t-anethole, germacrene-D, naphthalene, ledene, eucalyptol, azulene and comphore	[35]
*Mentha piperita*	Terpenes	GC-MS	Menthone, menthol, pulegone and menthofuran	[36]
*M.* *arvensis*	Terpenes	GLC	Menthol, isomenthone, L-methone and menthyl acetate	[37]
*Achyranthes bidentata*	Terpenes	HPLC	Oleanolic acid and ecdysterone	[38]
*Arabidopsis* *thaliana*	Phenolics	UHPLC-MS	Scopoletin, umbelliferone and esculetin, scopolin, skimmin and esculin	[39]
*P. ovata*	Phenolics	LC-MS	Luteolin, quercetagetin, syringetin, kaempferol, limocitrin, helilupolone and catechin	[10]
*P. ovata*	Phenolics	LC-MS	Kaempferol 3-(2″,3″-diacetylrhamnoside)-7-rhamnoside and apigenin 7-rhamnoside	[9]
*P. ovata*	Alkaloids	LC-MS	Lunamarine, hordatine B and pinidine	[10]
*Dendrobium* Snowflake ‘Red Star’	Alkaloids	^1^H and 2D NMR	Dendrobine and nobilonine	[40]

GC, gas chromatography; GC-MS, gas chromatography-mass spectrometry; GC-QqQ-MS, gas chromatography-triple quadrupole-mass spectrometry; GLC, Gas liquid chromatography; HPLC, high-performance liquid chromatography; HPLC-MS, high-performance liquid chromatography-mass spectrometry; LC-MS, liquid chromatography-mass spectrometry; ^1^H-NMR, nuclear magnetic resonance; UHPLC-MS, ultra-high performance liquid chromatography-mass spectrometry.

### 1.2. Secondary Metabolites

Secondary metabolites (SMs) play a crucial role in protecting plants against various environmental stresses. It has been estimated that approximately 100,000 SMs have been reported within different plant species and are classified into multiple groups, nitrogen-containing compounds, terpenes, thiols, and phenolic compounds [41]. In *Scutellaria baicalensis*, the major flavonoids are accumulated in the roots before the full-bloom stage [42]. Two flavonoids, kaempferol 3-(2″,3″-diacetylrhamnoside)-7-rhamnoside and apigenin 7-rhamnoside were found in all developmental stages of *P*. *ovata* [9]. The root of *Achyranthes bidentata*, oleanolic acid and ecdysterone levels are increased during the vegetative growth than in reproductive growth [38]. Nutraceutical flavonoids; luteolin, quercetagetin, syringetin, kaempferol, limocitrin, helilupolone and catechin/epicatechin/pavetannin B2 and were identified in leaf extract, whereas alkaloids, lunamarine and hordatine B were identified in the seed extract and pinidine was detected in the husk extract [10]. The plant growth regulators gibberellic acid (GA), indole -3-acetic acid (IAA) and 6-Benzylaminopurine (BAP) show that the main terpenes (methyl chavicol and trans-anethole) and other terpenes (eucalyptol and azulene) undergo certain changes depending on the type of the treatment of plant growth regulators in *O*. *gratissimum* [35]. The application of growth regulators enhances the production of essential oils (menthone, menthol, pulegone, and menthofuran) in *Mentha piperita*, which is revealed to be rich in economically important terpenes [36]. The foliar application of triacontanol significantly increased the amount of active terpenes (menthol, L-methone, isomenthone, and menthyl acetate) in *Mentha arvensis* [37]. Lin et al. [43] conducted a phytochemical screening of *Pteris vittata* and identified four flavonoids: quercetin, kaempferol, kaempferol-3-O-D-glucopyranoside and rutin [43]. Scoploletin, umbelliferone and esculetin, as well as their glycosides scopolin, skimmin, and esculin were found in *Arabidopsis thaliana* [39] (Table 1).

## 2. Involvement of Metabolomics in Genetically Modified (GM) Crops

Metabolomic techniques are rapidly being used to analyze genetically modified organisms (GMOs), allowing for a broader and deeper understanding of composition of GMO than standard analytical methods. Metabolomics studies revealed that malic acid, sorbitol, asparagine, and gluconic acid levels increased in *O. sativa* cultivated at different time points. In addition, mannitol, sucrose, and glutamic acid had a significant increase in transgenic rice grains as compared to non-genetically modified rice [44]. Metabolic profiling was performed in *Solanum tuberosum* DREB1A transgenic lines rd29A::DREB1A (D163 and D164), a 35S::DREB1A (35S-3) line, and non-transgenic [45]. Increased levels of the glutathione metabolite, γ-aminobutyric acid (GABA), as well as accumulation of β-cyanoalanine, a byproduct of ethylene biosynthesis, were observed in the DREB1A transgenic lines [45] (Table 2).

Metabolomic profiling also demonstrated that introduction of the *cold and drought regulatory-protein encoding CORA-like gene* (*SbCDR*) from *S. brachiata* into tobacco could enhance salt and drought tolerance by increasing the stress related metabolites such as proline, threonine, valine, glyceric acid, fructose, 4-aminobutanoic acid, asparagine [50]. Overexpression of a native *UGPase2* gene induced several metabolites related to amino acid, phenolic glycosides such as asparagine, γ-amino-butyric acid, aspartic acid, glutamine, 5-oxo-proline, 2-methoxyhydroquinone-1-*O*-glucoside, 2-methoxyhydroquinone-4-*O*-glucoside, salicylic acid-2-*O*-glucoside, 2,5-dihydroxybenzoic acid-5-*O*-glucoside, salicin in transgenic *Populus* lines [52]. Overexpression of *GmDREB1* in *T.*
*aestivum* substantially impacts numerous metabolic pathways involved in the biosynthesis of amino acids [54]. Tryptophan, leucine phenylalanine, valine, and tyrosine were significantly changed [54]. Some urea cycle-related metabolites, such as adenosine, arginine, allantoin, citrulline, adenosine monophosphate (AMP), hypoxanthine, and guanine, were significantly changed in the transgenic *T. aestivum* line [54]. The combination of modern analytical methodologies and bioinformatics tools in metabolomics provides extensive metabolites data that helps to confirm the significant equivalency and incidence of unanticipated alterations caused by genetic transformation (Table 2).

## 3. Significance of Sample Preparation in Plant Metabolites

In plant metabolomics study, plant samples are harvested, stored, metabolites extraction and quantification, followed by data interpretation. Sample preparation is a key step in plant metabolomics as it significantly changes the quantity of the metabolites. Thus, considering all the factors, harvesting and storage of plant samples should be quick as to reduce the changes of biochemical reaction in the plant cells [56]. Inappropriate handling during the sample collection is the most likely source of bias in plant metabolomic studies [57]. Sample harvesting, storage, and extract preparation should ideally follow the Metabolomics Standards Initiative (MSI) to justify plant metabolomics studies [58].

### 3.1. Sample Harvesting and Storage

Commonly, four major steps are involved in plant metabolomics; harvesting, storage, extraction, and sample analysis (Figure 1). Plant sample harvesting must be carried out with caution, as the metabolome of the plant is sensitive to enzymatic reactions that can degrade different metabolites. In addition, metabolites vary with the different development stages, plant age, and time of sample harvesting [6]. Mostly, 10–100 mg of plant samples are required for each biological sample in metabolomics studies. Usually, immediately after harvesting, the plant samples are snap-frozen in liquid nitrogen to prevent metabolic changes. Similarly, various storage techniques, such as freeze-drying, oven-drying, and air-drying, are essential for the processing of metabolomics [57,59].

### 3.2. Sample Preparation

Sample preparation plays a key role in metabolomic study, as it includes the extraction of metabolites using different extraction methods (Figure 1). Among the extraction methods, quenching, mechanical and ultrasound extraction methods are promising in the metabolomic analysis [60]. In addition, high quality, yield and chemical versatility can be obtained by integrating ultrasound extraction method and mechanical grinding [61]. Apart from extraction methods, the choice of solvents is also crucial, as a single solvent cannot extract a variety of metabolites (e.g., polar or nonpolar). A wide variety of metabolites can be isolated using a solvent system composed of chloroform: methanol: water [62,63]. This solvent system is widely used for a wide variety of metabolites such as polar compounds, nonpolar compounds, and hydrophilic metabolites. Diverse solvent systems were reported for the plant metabolomics, such as extraction with pure methanol [64,65], the mixture of methanol: water [66], and methanol: methyl-tert-butyl-ether: water [67]. A specific solvent gradient extraction method was developed to recover almost all types of metabolites in a single protocol [68]. In addition, hot methanol (70% *v*/*v*) was used to extract phenolic compounds from *Brassica oleracea* using ultra-high-performance liquid chromatography–diode array detector–tandem mass spectrometry [69]. Various methods are used for sample preparation, such as microwave-assisted extraction [70], ultrasound-assisted extraction [71], Swiss rolling technique [72], and enzyme-assisted extraction [73].

Targeted metabolite identification and quantification are the primary approaches for metabolomics investigation [74]. Sample preparation for target metabolites extracted from plant components such as leaves, stems, roots, etc., includes enrichment for metabolites of interest and removal of contaminants such as proteins and salts that hamper the analysis. Targeted metabolomics-based quantification aims for enhanced metabolite coverage by analyzing the selected metabolites [75]. The targeted metabolites extracted using different extraction methods such as different proportion of organic solvents [67], liquid–liquid extraction [75], and solid phase extraction method [76]. To increase analytical reliability, single or multiple internal standards can be spiked into the sample mixture during sample preparation [77]. In the final step of sample preparation for LC-MS, the solvents were evaporated, followed by re-dissolving the sample with a suitable solvent for LC-MS analysis [75]. Targeted metabolite quantification has been considered as the key method because of its reliable quantification accuracy, sensitivity and stability [78]. However, this method is typically confined to measuring a small number of known pre-selected analysts and is incapable of detecting unknown and novel metabolites. LC-multiple reaction monitoring (MRM)-MS approach has been employed for targeted metabolomics quantification analysis due to its rapid scan speed and good analytic stability [79]. New techniques have been developed to broaden the choices for targeted metabolomics research, using high-resolution equipment such as parallel reaction monitoring (PRM) [78]. In plant metabolomics, new extraction methods are also developing day by day in line depending on the nature of the compounds and selection of analytical systems.

## 4. Analytical Techniques Used for Plant Metabolome

Along with sample preparation, different MS-based analytical systems are available for data acquisition. In plant metabolomics, single analytical tools cannot be used to identify all the metabolites present in a sample; instead, a set of various techniques are needed to provide the largest amount of metabolite coverage [1]. Various metabolomics tools include MS-based techniques, namely GC-MS, LC-MS CE-MS, FTICR-MS MALDI, IMS, and NMR for sensitive and specific qualitative and quantitative analyses of metabolites (Figure 1) [6,80]. All seven mentioned analytical methods identifying metabolites in plant tissue directly or indirectly have advantages and disadvantages (Table 3). Also, the combination of analytical methods can be used to ensure the efficacy of metabolite profiling.

### 4.1. Gas Chromatography-Mass Spectrometry (GC-MS)

GC-MS is an ideal technique for the identification and quantification of small metabolites (~500 Daltons). These molecules include amino acids, fatty acids, hydroxyl acids, alcohols, sugars, sterols, and amines, which are identified mostly using chemical derivatization to make them volatile enough for gas chromatography [81]. Moreover, different methods of derivatization, such as alkylation, acylation, methoximation, trimethylsilylation, and silylation, can also be used. Two derivatization steps are required for the extraction and identification of metabolites using GC-MS. The first step requires the conversion of all the carbonyl groups using methoxyamine hydrochloride into corresponding oximes. The seconnhd step is followed by a trimethylsilylation reaction to increasing the volatility of the derivative metabolites using derivatizing reagents such as N-Methyl-N-(trimethylsilyl) trifluoroacetamide (MSTFA) and N,O-bis-(trimethylsilyl)-trifluoroacetamide (BSTFA) [82,83,84]. In this procedure, the hydrogen is replaced from the -NH, -SH, -OH and -COOH of specific metabolites with [-Si(CH3)3] and are converted into thermally stable, less polar and volatile trimethylsilyl (TMS)-ether, TMS-ester, TMS-amine, or TMS-sulphide groups, respectively [83]. Also, GC-MS is the preferable chromatographic technique for identifying low molecular weight compounds that are either volatile or can be converted into volatile and thermally stable metabolites by chemical derivatization prior to analysis [85]. The technique includes primary metabolites such as sugars, fatty acids, amino acids, long-chain alcohols, amines, organic acids, and sterols.

There are two major forms of ionization used in GC-MS that comprises of electron ionization (EI) and chemical ionization (CI). Till now, the majority of GC-MS methods in metabolomics utilize EI. GC with EI detector equipped with single quadrupole (Q) mass analyzer is the oldest and most advanced analytical tool with robustness, high sensitivity, resolution and reproducibility, but suffers from sluggish scanning speeds and also poor mass accuracy (~50–200 ppm). Therefore, GC with a time-of-flight mass spectrometry (TOF-MS) analyzer is more preferred for metabolic profiling as it provides higher mass accuracy, faster acquisition times, and improved deconvolution for complex mixtures [86]. Among all metabolomics techniques, GC-MS is one of the most standardized, efficient, productive technique in plant metabolomics and it is considered a most versatile platform for metabolites analysis [87]. In addition, GC-MS has the availability of the huge number of well-established libraries of both commercial and in-house metabolite databases [88,89,90]. Metabolite profiling is utilized as an essential tool for screening of GM crops with regard to quality and health requirements and in categorization to an investigation of potential changes in metabolic contents, e.g., *T. aestivum* [53], *O. sativa* [44], and *Z*. *mays* [91].

### 4.2. Liquid Chromatography-Mass Spectrometry (LC-MS)

LC-MS is one of the most comprehensive analytical techniques in plant metabolome research, which is used to measure a wide variety of complex metabolites. The LC-MS approach is appropriate for high molecular weight (>500 kDa) plant metabolites, heat-labile functional groups, chemically unstable functional groups, and high-vapor-point. It does not require volatilization of the metabolites. LC-MS is also quite effective techniques in profiling of SMs (e.g., alkaloids, phenolics, flavonoids and terpenes), lipids (e.g., phospholipids, sphingolipids and glycerolipids) and sterols, and steroids [19,24,92,93].

LC-MS can also be used with various ionization methods and depending on the choice of specific separating columns based on the chemical characteristics of both mobile and stationary phases [94]. Currently, reverse-phase columns such as C18 or C8 are the most widely used columns for LC gradient separation. In reverse-phase separations, organic solvent/aqueous mixed mobile phases are often used, such as water: acetonitrile or water: methanol. Atmospheric pressure ionization (API) and electron spray ionization (ESI) are the most widely used ionization tools for LC-MS [94,95]. ESI and API have provided limited structural information of the compound because they introduce less internal energy and produce only a few fragments [95]. Structural information is typically obtained by number of fragments using collision-induced dissociation (CID) on tandem MS^n^. Commonly, two tandem MS^n^ analytical tool configurations are commonly available with the LC-MS-based metabolite analysis: tandem-in-time and tandem-in-space. The ion trap MS is used by tandem-in-time instruments, such as quadruple ion traps (QIT-MS), FTICR-MS and orbitrap. The tandem-in-space tool facilitates two sequential steps of mass spectrometric analysis (MS2); it includes two mass analyzers separated by a collision cell [96,97]. Although LC-MS requires standard reference compounds to identify and quantify SMs, this restricts the analysis of metabolites that are not commercially available [98,99].

### 4.3. Capillary Electrophoresis-Mass Spectrometry (CE-MS)

CE-MS is a strong analytical technique for evaluating a large variety of ionic metabolites based on the proportion of charge and size ratio [93]. It provides fast and high-resolution of charged compounds from small injection volumes and enables the metabolites characterization based on mass fragmentation [57]. The coverage of CE-MS metabolites majorly overlaps with GC-MS, but requires no derivatization, thus this technique save time and consumables. CE is performed in a fused silica capillary tube, the ends of which are dipped in buffer solutions and across which high voltages (20–30 kV) are employed [84]. Furthermore, CE has low sensitivity and reproducibility, poor migration time and lack of reference libraries; therefore, it is the least appropriate platform for studying metabolites from complex plant samples [100,101]. However, CE has some distinct rewards over other metabolomics tools; primarily the fact that it uses low volume of separation, which is especially appropriate for the study of plant metabolome [57,102].

### 4.4. Fourier Transform ion Cyclotron Resonance-Mass Spectrometry (FTICR-MS)

FTICR-MS provides the highest resolving power and mass accuracy among all kinds of mass spectrometry [103]. Its specific analytical features have made FTICR an important technique for proteomics and metabolomics. The ability of FTICR–MS to provide ultimate high resolution and high mass accuracy data is now frequently used as part of metabolomics procedures [84]. It’s also well compatible with multi-stage mass spectrometry (MSn) analyzers. However, the instrument associated with a high magnetic field, complex ion-ion interactions and high cost are major barriers to its widespread application and use in plant metabolomics studies [56].

### 4.5. Matrix-Assisted Laser Desorption/Ionization (MALDI)

Recently, the applications of MALDI-Mass Spectrometry Imaging (MSI) and other MSI tools use a non-target approach for the qualitative or quantitative imaging of a broad variety of metabolites [104]. In plants, many studies have used MALDI-MSI to assess the spatial distribution of lipids, sugars and other classes of metabolites from plant parts such as flowers, leaves and roots [105,106]. In addition, MALDI-MSI has permitted the simultaneous analysis of the distribution of many peptides and proteins actively from a plant tissue section. This method involves coating a thin film of a matrix comprising either sinapinic acid, α-Cyano-4-hydroxycinnamic acid (CHCA) and 2,5-dihdroxybenzoic acid (2,5-DHBA) on the tissue surface. At each stage, a laser beam is inserted across the matrix-coated tissue to obtain a mass spectrum. For protein/metabolites imaging, MALDI is the most used method of ionization, combined with a wide variety of different mass analyzers, namely ToF, ToF-ToF, QqToF (quadrupole time of flight), Fourier ICR transform (FT-ICR), and ion-trap (both linear and spherical). All of these have their own merits and have previously been addressed and reviewed [107]. Other different ionization techniques such as secondary ion mass spectrometry (SIMS), desorption electrospray ionization (DESI) and laser ablation electrospray ionization (LAESI) have been also investigated [108].

### 4.6. Ion Mobility Spectrometry (IMS)

Ion mobility spectrometry (IMS), which separates gas ions based on their size-to-charge ratio, has become a robust separation method. IMS has been widely employed in a variety of research fields ranging from environmental to pharmaceutical applications [109,110,111,112,113,114]. The use of ion mobility has gained significance in bioanalysis due to the potential improvement of the sensitivity and the ability of the technique to distinguish highly related molecules based on conformational differences of molecules [115]. The IMS-derived collision cross-section indicates the effective area for the interaction between a particular ion and gas through which it travels [116]. Initially, IMS was utilized largely as a stand-alone technique; however, in recent years, the IMS coupling with MS (IMS-MS) has developed rapidly into a robust and extensively used separation technique with applications in many fields across the biological sciences, including the glycosciences [117]. IMS-MS developed quickly into a ready-to-use technique that became commercially accessible, particularly for glycan analysis [118]. The biological applications of IMS-MS for biomolecules include the analysis of oligonucleotides carbohydrates, steroid, lipids, peptides, and proteins [119,120,121,122,123]. Furthermore, IMS-MS may be hyphenated with front-end liquid chromatography (LC) separation to increase peak capacity and separation capabilities [123]. LC–IMS-MS technique has numerous significant benefits over other technologies in terms of increased peak capacity, isomer separation, and metabolite identification [123,124].

IMS-MS derived collision cross-section (CCS) value is high reproducible characteristic of metabolite ion, allowing for metabolite identification [125]. Therefore, the most essential aspect of metabolite identification in IMS-MS is the curation of the CCS database. Many in silico CCS databases, such as LipidCCS [126], MetCCS [127], and ISiCLE [128], have been curated and include over one million CCS values. Zhou et al. [129] developed the ion mobility new CCS atlas, namely, AllCCS for metabolite annotation using known or unknown chemical structures [129]. The AllCCS atlas included a wide range of chemical structures with >5000 experimental CCS records and ~12 million predicted CCS values for >1.6 million chemical molecules [129]. McCullagh et al. [130] used the ^TW^CCSN_2_ library to screen the steviol glycosides in 55 food commodities. Schroeder et al. [131] identified 146 plant natural compounds, 343 CCS values, and 29 isomers annotated (various flavonoids and isoflavonoids) in *Medicago truncatula* based on CCS, retention time, accurate mass, and molecular formula. The combination of a large-scale CCS database and different MS/MS spectra will assist in the discovery of new metabolites.

### 4.7. Nuclear Magnetic Resonance (NMR)

NMR is another popular analytical tool for investigating the varied metabolome in plants, involving the structure, content, and purity of molecules in the sample. As a result, metabolic profiling can provide qualitative and quantitative data from biological extracts [132]. The basic principle of NMR-based metabolite identification is to capture the radio frequency electromagnetic radiations emitted by atomic nuclei that have an odd atomic number (^1^H) or an odd mass number (^13^C) when placed in a strong magnetic field. Because there is no requirement for chromatographic separation or sample derivatization, the use of NMR has grown dramatically in recent years [94,133,134]. Furthermore, easy sample preparation procedures and excellent repeatability, non-destructive nature enables high throughput and quick analysis in NMR metabolomics but has less sensitivity than MS [135,136]. NMR is pH sensitive, buffered solutions are usually needed to keep the pH stable. A combination of methanol and aqueous phosphate buffer (pH 6.0, 1:1 *v*/*v*) or ionic liquids such as 1-butyl-3-methylimidazolium chloride has been shown to be the most effective in providing a comprehensive overview of both primary and secondary metabolites [137]. ^1^H NMR is quick and easy, it has been the leading metabolites profiling technique, but it suffers from signal overlapping in the complex mixture of plant extracts during metabolites profiling. However, other advanced 2D NMR-based techniques includes two-dimensional (2D) ^1^H J-resolved NMR, heteronuclear single quantum coherence spectroscopy (HSQC), heteronuclearmultiple quantum coherence (HMBC), total correlation spectroscopy (TOCSY) and nuclear overhauser effect spectroscopy (NOESY) [137]. High-resolution magic angle spinning (HRMAS)-NMR is particularly well suited for solid lyophilized tissue without the need for chemical extraction, which is essential for both MS and liquid state NMR practices [86]. The acquisition time for 2D NMR (2D J-resolved spectroscopy) is around 20 min, whereas for one-dimensional (1D) NMR it is approximately 1 min. However, due to the dispersion of the resonance peaks in a second dimension, spectral overlapping can be reduced in 2D NMR J-resolved spectroscopy to detect signals in crowded spectral regions [138]. Using advanced NMR, glycine-betaine, citric acid, trehalose and ethanol levels were higher in *Cry1Ab* gene transformed maize plants than non-transgenic maize plants showed [55]. Transgenic maize plants showed lower levels of pyruvic, isobutyric, succinic, lactic, and fumaric acids than non-transgenics [55]. During seed germination in chickpea, the exogenous uptake of glucose in presence of nitric oxide donor was quantified by using ^1^H-NMR [33].

## 5. Metabolomic Data Processing, Annotation, Database and Bioinformatics Tools for Plants METABOLOME Analysis

GC-MS, LC-MS CE-MS, FTICR-MS MALDI, IMS and NMR are perhaps the most important techniques within the context of natural product discovery. Metabolomics generate a huge amount of metabolic data using wide range of analytical instruments. During the last decade, different software tools (web-based programs) have been designed for metabolomics raw data processing, data mining, data assessment, data interpretation, and statistical analysis as well as mathematical modelling of metabolomic networks (Figure 1).

### 5.1. Data Processing and Annotation

Several software programs are available for in silico data analysis of a large quantity of spectrum data of metabolites generated by various analytical instruments. The web-based programs were used for raw data processing, mining, and integration of metabolites. In general, acquired data is processed for the correction of baseline shifts, background noise reduction, peak detection and alignment, and finally, deconvolution of mass spectra (Figure 1, Table 4). Many bioinformatic tools are designed for the data pre-processing, including XCMS (https://xcmsonline.scripps.edu, accessed on 29 June 2021), METLIN (http://metlin.scripps.edu, accessed on 29 June 2021) AMDIS (Automated Mass Spectral Deconvolution and Identification System), MeltDB, MetaboAnalys, MetAlign, MZmine 2, and AnalyzerPro for different analytical techniques (Table 1). XCMS is an online bioinformatics platform that facilitates the direct uploading of raw data and assists the user in data processing and statistical analysis [139]. For LC-MS experiments, XCMS has been developed for programmed data transfer that has reduced data processing time and improved the effectiveness of an online system [140]. METLIN is another online database, which has been used in various studies related to plant metabolic profiling of stress response. It is useful for plant metabolic profiling of specific metabolites, and it is not time-consuming for data processing, mining, and annotation [141].

MeltDB (https://meltdb.cebitec.uni-bielefeld.de, accessed on 29 June 2021) is an important web-based platform used for data assessment, processing, and statistical analysis in plant metabolomics [142]. In addition, MetaboAnalyst online platform also includes a flexible enrichment analysis tool including some topological and visualization possibilities [143]. Global natural product social molecular networking (GNPS; http://gnps.ucsd.edu, accessed on 29 June 2021) is web-based mass spectrometry (MS/MS) for processing and annotation of metabolites [144]. GNPS assists with the identification and discovery of metabolites throughout the data, from data acquisition/analysis to post-publication [144]. Finally, the MZmine 2 is a publicly accessible data processing module that supports high-resolution spectral analysis. MZmine 2 is suitable for both targeted and non-targeted metabolomic studies, and it is well suited for processing large batches of data [145]. Various computational web-based, statistical and online bioinformatics tools are commonly used for data analysis in plant metabolomics (Table 4).

**Table 4 plants-10-02409-t004:** Available/accessible bioinformatics and statistical tools for metabolite identification.

Database Name	Website (URL, Accessed on 29 June 2021)	Data Input	Major Function	Reference
ADAP	http://www.du-lab.org/software.htm/	GC/TOF-MS	Data processing	[146]
AllCSS	http://allccs.zhulab.cn/	DTIM-MS TWIM-MS	Metabolite prediction and annotation	[129]
AMDIS	http://www.amdis.net/	GC-MS	Data processing	[147]
BinBase	http://fiehnlab.ucdavis.edu/db or https://fiehnlab.ucdavis.edu/projects/binbase-setup	GC-MS	Metabolite annotation	[148]
FiehnLib	http://fiehnlab.ucdavis.edu/db or https://fiehnlab.ucdavis.edu/projects/fiehnlib	GC-qTOF-MS	Metabolic profiling	[149]
GMDB	https://jcggdb.jp/rcmg/glycodb/Ms_ResultSearch	MALDI-TOF	Metabolite annotation	[150]
GNPS	https://gnps.ucsd.edu/ProteoSAFe/static/gnps-splash.jsp	GC-MS-EI LC-MS	Data processing, visualization and metabolite annotation	[144]
KEGG	http://www.genome.jp/kegg/	--	Metabolic models	[151]
KNApSAcK	http://kanaya.naist.jp/KNApSAcK/	FT/ICR-MS	Metabolite database	[152]
MarVis	http://marvis.gobics.de/	LC-MS	Metabolite annotation	[153]
MassBase	http://webs2.kazusa.or.jp/massbase/	MS	Metabolite annotation	[154]
MAVEN	https://maven.apache.org/	LC-MS	Data processing	[155]
MeltDB 2.0	https://meltdb.cebitec.uni-bielefeld.de	GC-MS & LC-MS	Data processing	[142]
MetaboAnalyst	www.metaboanalyst.ca/	GC-MS & LC-MS	Statistical analysis	[156]
Metabolome Express	https://www.metabolome-express.org	GC-MS	Data processing, visualization and statistical analysis	[157]
MetaboSearch	http://omics.georgetown.edu/metabosearch.html	MS	Data annotation	[158]
Metabox	https://github.com/kwanjeeraw/metabox	MS	Analysis workflow	[159]
MetAlign	www.metalign.nl	GC-MS & LC-MS	Data processing & Statistical analysis	[160]
metaP-server	http://metabolomics.helmholtz-muenchen.de/metap2/	LC-MS/MS	Data analysis	[161]
MetAssign	http://mzmatch.sourceforge.net/	LC-MS	Data annotation	[162]
MetFrag	https://ipb-halle.github.io/MetFrag/	MS	Metabolite annotation	[163]
MET-IDEA	http://bioinfo.noble.org/gateway/index.php?option=com_wrapper&Itemid=57	GC-MS & LC-MS	Data processing	[164]
MetiTree	http://www.metitree.nl/	MS	Data annotation	[165]
METLIN	https://metlin.scripps.edu/	LC-MS & MS/MS	Metabolite annotation	[141]
MMCD	http://mmcd.nmrfam.wisc.edu/ or https://www.g6g-softwaredirectory.com/bio/metabolomics/dbs-kbs/20670-Univ-Madison-WI-MMCD.php	MS	Metabolite annotation	[166]
Molfind	http://metabolomics.pharm.uconn.edu/Software.html	HPLC/MS	Metabolite annotation	[167]
Mzcloud	https://www.mzcloud.org/	MS/MS & MSn	Metabolite annotation	[168]
MZedDB	http://maltese.dbs.aber.ac.uk:8888/hrmet/index.html	MS	Data annotation	[169]
MZmine2	http://mzmine.github.io/	LC-MS	Data processing	[145]
NIST	http://www.nist.gov/srd/nist1a.cfm or https://www.nist.gov/srd/nist-standard-reference-database-1a	GC-MS, LC-MS & MS/MS	Metabolite annotation	[170]
PRIMe	http://prime.psc.riken.jp/	GC-MS, LC-MS & CE-MS	Metabolite annotation	[171]
XCMS	https://xcmsonline.scripps.edu	GC-MS, LC-MS & MS2	Data processing	[139]

CE-MS, capillary electrophoresis-mass spectrometry; DTIM-MS, drift tube ion mobility–mass spectrometry; EI, electrospray ionization; FTICR-MS, fourier transform ion cyclotron resonance-mass spectrometry; GC-TOF-MS, gas chromatography-time of flight-mass spectrometry; GC-MS, gas chromatography-mass spectrometry; HPLC, high-performance liquid chromatography; LC-MS, liquid chromatography-mass spectrometry; MALDI-TOF, matrix-assisted laser desorption/ionization- time of flight; TWIM-MS, traveling wave ion mobility–mass spectrometry.

### 5.2. Network Analysis

The basic goal of pathway analysis is to combine biochemical information with collected metabolomics data to recognize metabolite patterns that match with metabolic pathways [172]. It is possible to consider metabolic pathways as groups of metabolites that share a common biological process and are related by one or more enzymatic reactions. A broad set of metabolic pathways are covered by comprehensive metabolic pathway databases, such as the KEGG database [173], MetaCyc [174], AraCyc [175] and the small molecule pathway database (SMPDB) [176] (Table 5). A number of software, such as, metabolite set enrichment analysis (MSEA), MPEA, IMPaLA, MBRole, VANTED, MetaboAnalyst, Paintomics, ProMeTra, Metscape2, and MetaMapRR can perform statistical and other metabolite enrichment analyses (Table 5). MSEA methods can be methodically distinguished into over-representation (ORA), single-sample profiling (SSP) and quantitative enrichment (QEA) analysis [177]. Metscape2 [178], which is an add-on to the common Cytoscape software [179] that allows data on metabolites, genes, and pathways to be displayed in the scope of metabolic networks. In addition, platform-independent online resources such as Paintomics [180], ProMeTra [181] and MetaMapRR [182] are also accessible.

## 6. Conclusions

Metabolomics has achieved a prominent role in plant science research. It has wide applications ranging from investigating the stress-specific metabolites for different climatic stresses, evaluating candidate metabolic gene functions to analyzing the biological mechanism in plant cells, and dissecting the genotype-phenotype relationship in response to the various biotic and abiotic stresses. This review provides an overview of different sample collection, harvesting methods, storage, and sample preparation in the plant metabolomics experiments. Furthermore, the most widely used analytical tools in metabolomics for agriculture research viz. GC-MS, LC-MS, CE-MS, FTICR-MS, MALDI, IMS, and NMR with new development in their applications. In addition, we discussed computational software and database employed for metabolomics data processing in plant science. The integration of comprehensive bioinformatics tools with omics strategies professionally dissects novel metabolic networks for crop improvement. Metabolomics has excelled classical approach for novel metabolites discovery and simultaneously explores the complexity and enormous chemical diversity of metabolites in any crop plant. The integration of metabolomics with other “omics” technologies, e.g., genomics, transcriptomics, proteomics, can deliver novel insights into crop plants’ genetic regulations in the context of their cellular function and metabolic network. The complete elucidation of physio-biochemical and molecular mechanisms underlying plant developmental and stress-responsive biology primarily depends on the comprehensive investigations using omics techniques that make metabolomics more applicable in agriculture sciences. Metabolomics has tremendous potential in plant research, as metabolites are more appropriate to the plant phenotype than DNAs, RNAs, or proteins. Therefore, studies in this field will effort on both ways, one is the systematic study of the biochemical and genetic mechanisms of metabolic variations in crop plants using both targeted and non-targeted methods; other is metabolomic platform can be used for metabolic profiling of genome-edited plants using CRISPR/Cas9 system for risk evaluation and regulatory affairs related with genetically modified crops [196]. Thus, we can say metabolomics will be able to contribute a lot to agriculture science, such as crop breeding and genome editing for crop improvement, better grain yield, and elucidating their unknown and novel metabolic pathways.

## Figures and Tables

**Figure 1 plants-10-02409-f001:**
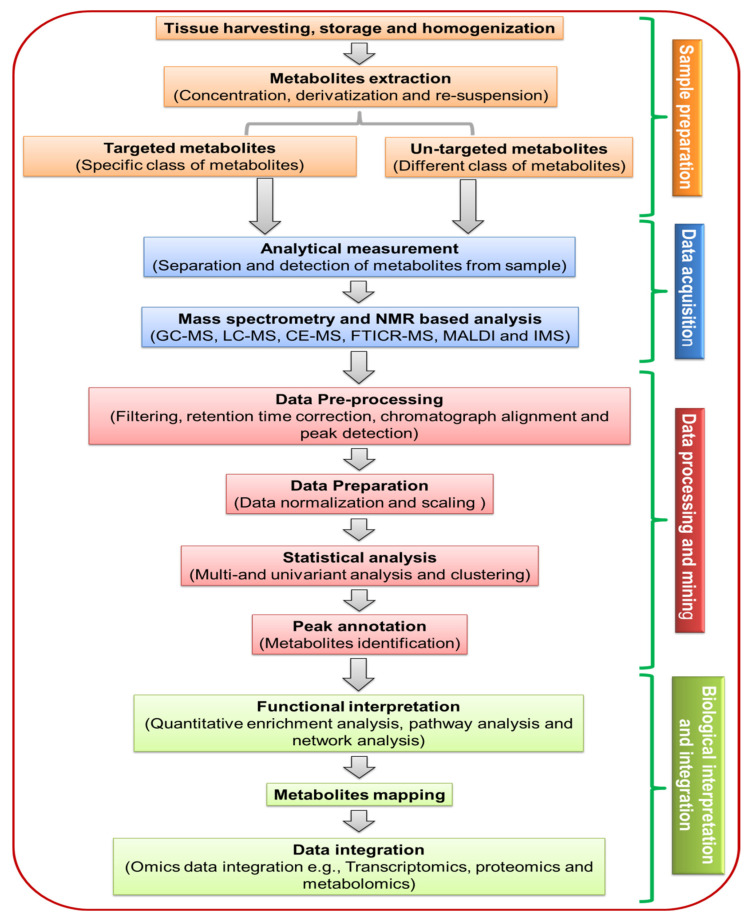
Schematic representation of the multi-step workflow of a plant metabolomics study. Sample preparation, data acquisition, data processing and biological interpretation are key steps in plant metabolomics. Nowadays, for data acquisition, different MS-based analytical tools (GC-MS, LC-MS CE-MS, FTICR-MS, MALDI, and IMS) and NMR are available. The most important step in data processing and mining includes correction of baseline shifts, background noise reduction, chromatograph alignment and peaks detection. Biological interpretation and integration include enrichment analysis, networks, and pathways analysis for a comprehensive scope of the metabolome. GC-MS, gas chromatography-mass spectrometry; IMS, ion mobility spectrometry; LC-MS, liquid chromatography mass-spectroscopy; CE-MS, capillary electrophoresis-mass spectrometry; FTICR-MS, fourier transform ion cyclotron resonance-mass spectrometry; MALDI, matrix-assisted laser desorption/ionization; NMR, nuclear magnetic resonance.

**Table 2 plants-10-02409-t002:** Identification of important metabolites in transgenic plants using different analytical tools.

Transgenic Plants	Analytical Techniques	Key Metabolites	References
*Artemisia annua*	GC-TOF-MS	Borneol, phytol, β-farnesene, germacrene D, artemisinic acid, dihydroartemisinic acid, and artemisinin	[46]
*Lactuca* *sativa*	NMR	Asparagine, glutamine, valine, isoleucine, α-chetoglutarate, succinate, fumarate, malate, sucrose, and fructose	[47]
*Lycopersicon esculentum*	GC-MS	γ-aminobutyric acid, histidine, proline, pyrrol-2-carboxylate, galactitiol/sorbitol, glycerol, maltitol, 3-phosphoglyceric acid, allantoin, homo-cystine, caffeate, gluconate, ribonate, lysine, threonine, homo-serine, tyrosine, tryptophan, leucine, arginine and valine	[48]
*Nicotiana tabacum*	NMR	Chlorogenic acid, 4-O-caffeoylquinic acid, malic acid, threonine, alanine, glycine, fructose, β-glucose, α-glucose, sucrose, fumaric acid and salicylic acid	[49]
*N. tabacum*	GC-MS	4-Aminobutanoic acid, asparagine, glutamine, glycine, leucine, phenylalanine, proline, serine, threonine, tryptophan, chlorogenic acid, quininic acid, threonic acid, citric acid, malic acid and ethanolamine	[50]
*Oryza sativa*	GC-MS	Glycerol-3-phosphate, citric acid, linoleic acid, oleic acid, hexadecanoic acid, 2,3-dihydroxypropyl ester, sucrose, 9-octadecenoic acid, 2,3-dihydroxypropyl ester, sucrose, mannitol and glutamic acid	[44]
*O. sativa*	LC-MS	Tryptophan, phytosphingosine, palmitic acid, 5-hydroxy-2-octadenoic acid 9,10,13-trihydroxyoctadec-11-enoic acid and ethanolamine	[51]
*Populus*	GC-MS, HPLC	Caffeoyl and feruloyl conjugates, syringyl-to-guaiacyl ratio, asparagine, glutamine, aspartic acid, γ-amino-butyric acid, 5-oxo-proline, salicylic acid-2-O-glucoside, 2, 5-dihydroxybenzoic acid-5-O-glucoside, 2-methoxyhydroquinone-1-O-glucoside, 2-methoxyhydroquinone-4-O-glucoside, salicin, gallic acid, and dihydroxybenzoic acid	[52]
*Solanum tuberosum*	LC-TOF-MS	Glutathione, γ-aminobutyric acid, β-cyanoalanine, 5-oxoproline, sucrose, glucose-1-phosphate, glucose-6-phosphate, fructose-6-phosphate, ethanolamine, adenosine, and guanosine	[45]
*Triticum aestivum*	GC-MS	Guanine and 4-hydroxycinnamic acid	[53]
*T. aestivum*	LC-MS	Aminoacyl-tRNA biosynthesis, phenylalanine, tyrosine, tryptophan glyoxylic, tartaric acid, oxalic acids, sucrose, galactose, mannitol, leucine, valine, glutamate, proline, pyridoxamine, glutathione, arginine, citrulline, adenosine, hypoxanthine, allantoin, and adenosine monophosphate	[54]
*Zea mays*	^1^H NMR	Lactic acid, citric acid, lysine, arginine, glycine-betaine, raffinose, trehalose, galactose, and adenine	[55]

GC-MS, gas chromatography-mass spectrometry; GC-TOF-MS, gas chromatography-time of flight-mass spectrometry; HPLC, high-performance liquid chromatography; LC-MS, liquid chromatography-mass spectrometry; LC-TOF-MS, liquid chromatography-time of flight-mass spectrometry; ^1^H-NMR, nuclear magnetic resonance.

**Table 3 plants-10-02409-t003:** Advantages and disadvantages of common analytical techniques used in MS-based and NMR metabolomics.

Analytical Method	Advantage	Disadvantage
**GC-MS**	Suitable for the identification of thermally stable and volatile compounds Large commercial and public libraries Identification of low molecular weight metabolites (~500 daltons)	Sample pre-processing process and requires derivatization Many metabolites are thermally unstable or unsuitable for non-volatile compounds
**LC-MS**	Easy sample preparation No derivatization Several separation modes are available Multiple MS detectors Large number of detectable metabolites	Few commercial libraries Adduct ions are needed for metabolites detection
**CE-MS**	Evaluating ionic metabolites based on the proportion of charge and size ratio Fast and high-resolution of charged compounds No derivatization	Low sensitivity and reproducibility Poor migration time and lack of reference libraries
**FTICR-MS**	Mass resolving power Mass accuracy and dynamic range	Expensive Lack of detection for non-ionizable compounds Slow MS/MS
**MALDI-MSI**	Quantification by peak intensities Resolution up to 10 µm Direct on tissue identification by tandem–MS fragmentation Mass range up to 20 kDa	Unsuitable for higher molecular mass compound Expensive equipment to purchase Time consuming Limited by size of the metabolites
**IMS**	Ion fragmentation with high versatility Gold standard CCS values High resolution; IMS^n^ (Ion mobility spectrometry)	Low ion mobility resolution Resolution depends on the number of passes Mass range depends on ion mobility resolution
**NMR**	Precise quantification and reproducibility Simple steps of sample preparation Separation is not required. Provide detailed information about the structure of known and undiscovered metabolite Acceptable with liquids and solids samples	Expensive cost of instrument Low sensitivity Inadequate bioinformatics platform A large amount of sample is required. Spectral analysis is a tough and time-consuming process.

**Table 5 plants-10-02409-t005:** Database for metabolite enrichment analysis and pathway visualization.

Database	Website (URL, Accessed on 29 June 2021)	References
AraCyc	https://www.plantcyc.org/typeofpublication/aracyc	[175]
Cytoscape	http://www.cytoscape.org/	[183]
IMPaLA	http://impala.molgen.mpg.de	[184]
iPath	http://pathways.embl.de/	[185]
KEGG	http://www.genome.jp/kegg/	[173]
MapMan	http://mapman.gabipd.org/web/guest/mapman	[186]
MBRole	http://csbg.cnb.csic.es/mbrole/	[187]
Metabolonote	http://metabolonote.kazusa.or.jp/	[188]
MetaCrop	http://metacrop.ipk-gatersleben.de	[189]
MetaCyc	http://www.metacyc.org	[174]
MetPA	http://metpa.metabolomics.ca/MetPA/	[190]
MPEA	http://ekhidna.biocenter.helsinki.fi/poxo/mpea/	[191]
MSEA	http://www.msea.ca. or http://www.metaboanalyst.ca	[177]
Pathcase	http://nashua.case.edu/PathwaysMAW/Web/	[192]
PathwayExplorer	http://genome.tugraz.at/pathwayexplorer/pathwayexplorer_description.shtml	[193]
SMPDB	http://www.smpdb.ca	[176]
VANTED	https://immersive-nalytics.infotech.monash.edu/vanted/	[194]
WikiPathways	http://wikipathways.org	[195]

## Data Availability

All data included in the main text.

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
