# Peer review of "Plants Metabolome Study: Emerging Tools and Techniques"

_plants, 2021, doi:10.3390/plants10112409_

Round 1

Reviewer 1 Report

I have completed the review of the article: Plants metabolome study: emerging tools and techniques. This review about the techniques used in the study of plant matabolomics is very topical, the study is well organized and presented. I have only one suggestion, namely to include several concrete examples in which various techniques can be used in metabolomics studies, as it is very well and concretely presented the use of these techniques in the study of genetically modified plants (L177).

Reviewer 2 Report

Dear Authors,

The manuscript needs profound improvements, please see attached pdf file for my comments and remarks.

Additionaly, in my opinion the manuscript requires improvements concerned to following problems:

  1. Strategy description during targeted metabolites profiling - samples preparation and isolation of target metabolites.
  2. Primary and secondary metabolites (their classes).
  3. Something about plant lipidomics should be added.
  4. In the present form it can not be published.

Reviewer 3 Report

The review entitled “Plant metabolome study: emerging tools and techniques” by Patel and co-workers proposes to summarize the main analytical platforms and bioinformatics tools employed in plant metabolomics. Although the work is well-written, it requires major revisions in order to be accept for publication.

Major comments

Data Acquisition. There is no mention about NMR spectroscopy. This analytical method has been employed in plant metabolomics for at least 20 years, starting with the work developed by Hostettmann and Verpoorte. A review that purports to present emerging technologies needs to point to trends in this area. Is this technology no longer used in plant metabolomics? If it is still employ, in what way?
The description of the main analytical approaches using mass spectrometry seems adequate, with exception of GC-MS. This technique is used for volatile and non-polar compounds, and polar compounds after derivatization. The authors need to add the analysis of volatile and non-polar compounds in the GC-MS section and better explain in the table. I also missed recent advances in the field, such as the application of ion mobility coupled with MS (IMS-MS).

Bioinformatics. The section needs to be break into pre-processing approaches and metabolite annotation/analysis. Then, the authors need to better explain the main options and how they proceed. It is important to add mzMine 2.0 in the section. It is also important to describe how to convert raw data in generic file formats to use in this software. In the metabolite annotation/analysis section, the authors need to describe GNPS platform. This is one of the most important online tool for data visualization and metabolite annotation in the field of natural products, including plants.

Examples. The authors should also provide a table containing the classes of metabolites that can be measured for each analytical technique.

Round 2

Reviewer 2 Report

Dear Authors,

In my opinion, the new version of the manuscript contains appropriate supplements, which makes it possible to publish this review article.

Reviewer 3 Report

The authors should revised table 1. I was not able to find the key metabolites pointed to ref 33. Besides, I suppose fatty acid methyl esters are internal standards for GC-MS analysis, not metabolites extracted from the plant.

GNPS does not work exclusively with GC-MS-EI, but also LCMS
